# Vertical tearing of subducting plates controlled by geometry and rheology of oceanic plates

Yaguang Chen [1,2], Hanlin Chen [1] ✉, Mingqi Liu [2] ✉ & Taras Gerya [2] ✉

Lateral non-uniform subduction is impacted by continuous plate segmentation owing to vertical tearing of the subducting plate. However, the dynamics and physical controls of vertical tearing remain controversial. Here, we employed 3D numerical models to investigate the effects of trench geometry (offset by a transform boundary) and plate rheology (plate age and the magnitude of brittle/plastic strain weakening) on the evolution of shear stress-controlled vertical tearing within a homogenous subducting oceanic plate. Numerical results suggest that the trench offset geometry could result in self-sustained vertical tearing as a narrow shear zone within the intact subducting oceanic plate, and that this process of tearing could operate throughout the entire subduction process. Further, the critical trench offset length for the maturation of vertical tearing is impacted by plate rheology. Comparison between numerical modelling results and natural observations suggests that vertical tearing attributed to trench offset geometry is broadly developed in modern subduction and collision systems worldwide.

Vertical tearing (VT) is a lithospheric rupture of the subducting plate that propagates sub-parallel and opposite to the subduction direction[1-3]. In terms of the fracture mechanism, VT can be divided into mode-I (tension) and mode-III (out-of-plane shear) based on the crack surface displacement[4] (Fig. 1a). The mode-I VTs require some specific dynamic processes to generate the subduction-perpendicular extensional stress, e.g., active mid-ocean ridge subduction[5], the difference of subducting direction[6], or flattening of a curved subducted slab[7]. Comparatively, the shear stress-controlled mode-III VT is a more prevalent type in both subduction and collision zones[8,9] (Fig. 1b), as accommodating lateral non-uniform plate motions and controlling the segmentation of subducting plates to enable continuous subduction[2]. The generalized mode-III VT can be detected in different geodynamic settings, within two subducting segments[2,10-13] (the mode-III VT, which separates two offset subducting plate sections) or near the terminations of subduction zones[1,14] (the STEP, i.e., Subduction-Transform Edge Propagator[1], which separates subducting and non-subducting plates) (Fig. 1b).

The lateral non-uniform subduction associated mode-III VT can be triggered by the stress heterogeneity resulting from the lateral variation of physical properties (e.g., plate age[15], plate strength[16], or subduction rate[2], et al.). However, both analog and numerical modeling studies have shown that these physical properties are insufficient to segment the intact subducting plate[17,18]. Hence, pre-existing lithospheric anomalies (buoyant terranes[19,20] or weak zones combined with lateral heterogeneity[3,15]) are usually considered to induce the mode-III VT. But these pre-existing lithospheric anomalies seem unnecessary in the tearing propagation stage. The lack of pre-existing lithospheric anomaly around the active mode-III VTs in the Aleutian–Kamchatka trench[10,21], the Solomon-Vanuatu trench[12], the southeastern Aegean[11], and et al. (Fig. 1b) indicates that the mode-III VT can propagate within the homogenous subducting plate without the participation of lithospheric anomalies. Thus, a new factor is needed to dominate the propagation of mode-III VTs within the complete subducting plate.

Both mode-III VTs and STEPs are generally formed and propagating around the transition regions between subduction zones and

[1]Key Laboratory of Geoscience Big Data and Deep Resource of Zhejiang Province, School of Earth Sciences, Zhejiang University, Hangzhou, China. [2]Department of Earth Sciences, Institute of Geophysics, ETH Zürich, Zürich, Switzerland. ✉e-mail: hlchen@zju.edu.cn; ml_013@usc.edu; taras.gerya@erdw.ethz.ch

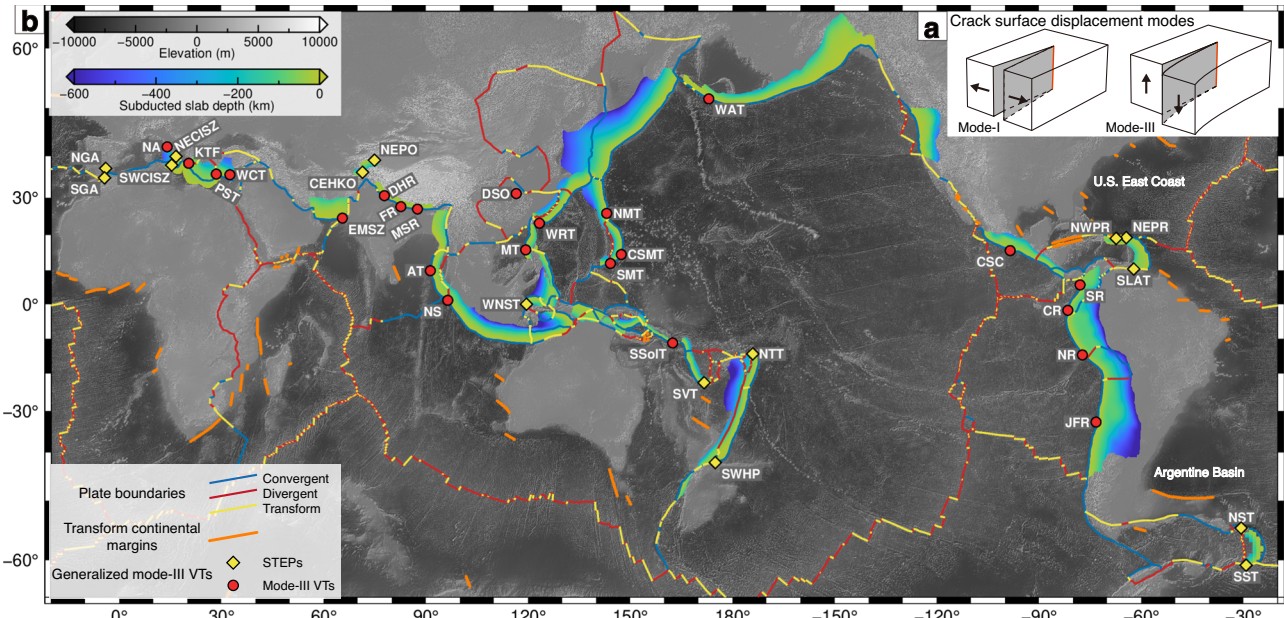

**Fig. 1 | The global distribution of proposed generalized mode-III VTs. a** The crack surface displacement modes[4]. Mode-I refers to opening or tension mode, where the crack surfaces (gray planes) move directly apart in a direction perpendicular to themselves. Mode-III refers to out-of-plane shear mode, where the crack surfaces move relative to one another in a direction paralleling the crack tip (red line). **b** The generalized Mode-III VTs. Based on their geodynamic settings, they are subdivided into STEPs (yellow diamonds, near the subduction zone edge) and mode-III VTs (red cycles, within two subducting segments). The topography data is from ETOPO1 arc-minute Global Relief Model[75]. The slab depth shows the geometry of subducting slabs[12]. The color of the lines shows the different plate boundary types from Bird (2003)[13]. The blue lines indicate convergent boundaries, consisting of subduction zones and oceanic and continental convergent boundaries. The red lines indicate divergent boundaries, including oceanic spreading ridges and continental rift boundaries. The yellow lines indicate the transform boundaries, containing oceanic and continental transform faults. Transform continental margins from Mercier de Lépinay et al. (2016)[14] are marked by orange lines. Abbreviation descriptions and approximate locations of VTs are summarized in Supplementary Table 1.

transform boundaries (Fig. 1b). The STEP, developing near the edges of subduction zones, has been proposed as a trench geometric consequence[1]. The mode-III VT is characterized by a similar trench geometry, where the intact trench is offset into two sections accompanied by the continuously growing transform boundary (strike-slip faults). The potential relationship between this trench offset geometry and mode-III VT has been suggested in the previous study, where trench offset geometry may cause a volume reduction of the subducting plate that can be accommodated by the mode-III VT[22]. Moreover, this trench offset geometry is more prominent in regions where the mode-III VTs propagate within the intact subducting plate (as examples mentioned above) than those where mode-III VTs controlled by the pre-existing lithospheric anomalies (e.g., the subduction of the fracture zone in the Central-Southern Cocos[23] and aseismic ridge subduction beneath the Andes[24–26]), implying that, within a complete subducting plate, the propagation of mode-III VT may be more closely related to the trench offset geometry. However, it remains uncertain, within a homogenous subducting plate, what the exact controlling relation between the trench offset geometry and mode-III VT is, which physical properties control the dynamics of mode-III VT, and what characteristics the mode-III VT has. Linking the mode-III VT and the trench offset geometry would be an excellent point to understand these questions.

To investigate the dynamics, characteristics, and physical controls of a mode-III VT within an intact subducting oceanic plate, we employed 3D high-resolution visco-plastic thermomechanical numerical models containing subduction zones initiated from the initially offset trench geometry corresponding to the transform boundary (Supplementary Fig. 1a). We systematically test the effects of the trench offset length and subducting plate rheology on the maturation and propagation of mode-III VT. The plate rheology parameters contain the plate age and the magnitude of brittle/plastic

strain weakening, where the strain weakening is realized by varying the internal friction coefficient of the lithospheric mantle (Methods). The model results reflect that mode-III VT is a self-sustained lithospheric narrow shear zone that can propagate infinitely if the condition does not change. The long-term stability of mode-III VT is critically controlled by the trench offset geometry, and the required trench offset length for a mature VT is strongly affected by the lithospheric rheology. Furthermore, by comparing the STEP and the mode-III VT, we note that they have different geodynamic settings but share the same tear propagation mechanisms and features. Formally, the STEP can be regarded as a mode-III VT with an infinite long trench offset.

## Results and discussion
### Reference model evolution
The typical model evolution shows the maturation and stable propagation of a self-sustained mode-III VT within an intact subducting plate resulting from the trench offset geometry (Supplementary Fig. 2). At the initial stages, stress concentrates at the outer subduction-transform corner under a constant far-field convergence velocity (Supplementary Fig. 2a). This stress field heterogeneity originates from the trench offset geometry that amplifies the bending and intra-plate shear stresses related to subduction initiation. After -0.5 Ma, the oceanic plate begins to bend down gradually (Supplementary Fig. 2b). An incipient vertical shear zone develops due to plastic blunting[27] in response to increasing local stress concentration. After 1.2 Ma, the weakened area tends to be sharper and narrower, leading to the growth of the vertical shear zone with an orientation opposite to the subduction direction (Supplementary Fig. 2a). This process is attributed to the strain localization process, which is facilitated by the brittle/plastic strain weakening of the plate (Methods). The deformation rate of mode-III VT accelerates rapidly as a consequence of strain

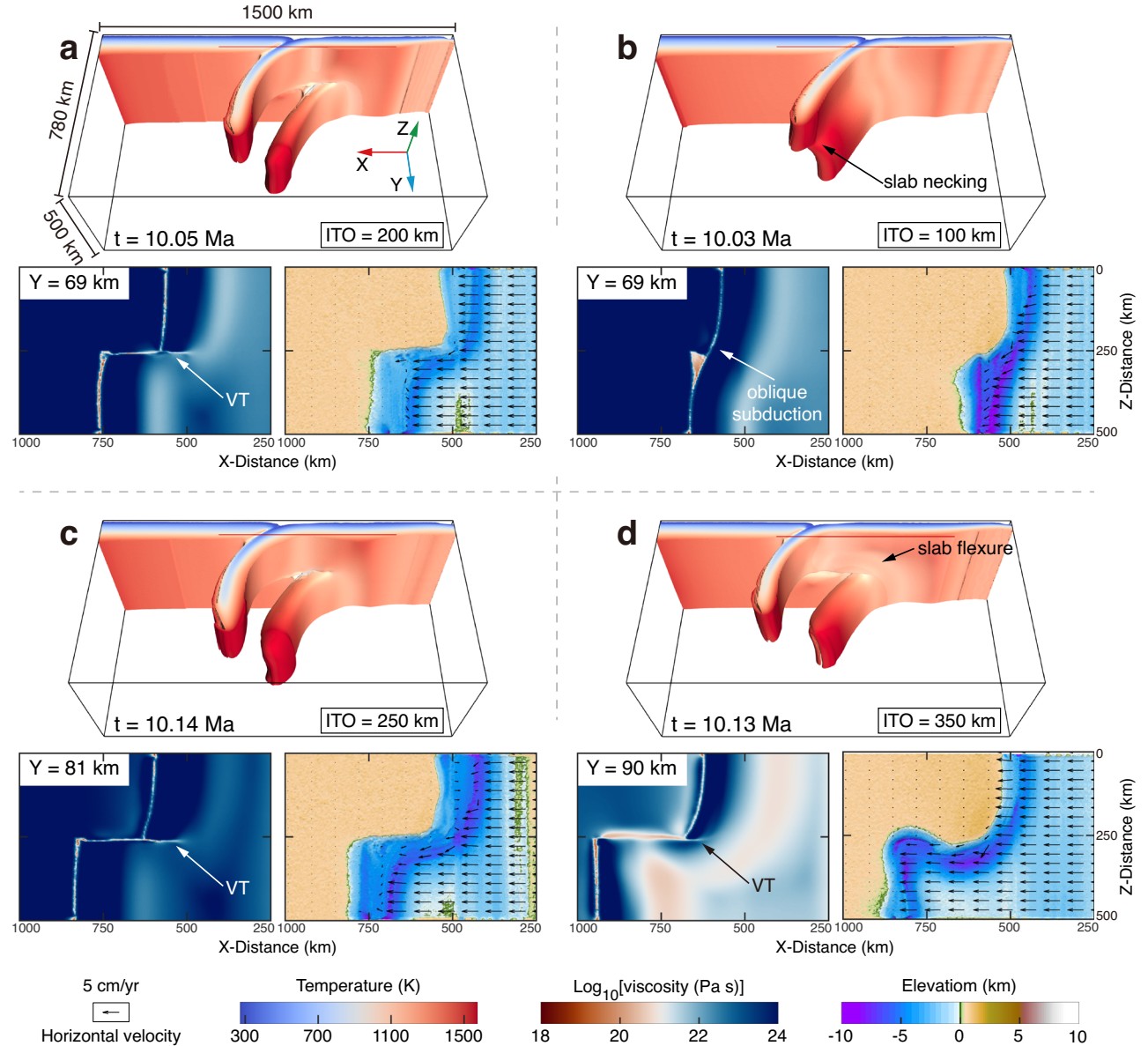

**Fig. 2 | Variations of model results for different parameters. a** Reference model (Mod12). **b**–**d** are the same as the reference model, but 100 km trench offset (**b**, Mod10), 80 Ma oceanic plate age (**c**, Mod29), or no strain weakening (**d**, Mod55) is applied. In each section, subducting plate morphology is shown at the top (by the $10^{22}$ Pa s iso-viscosity contour); the horizontal viscosity slice is at the bottom left, marked by the red line and transparent surface in slab morphology; relative surface elevation is at the bottom right superimposed by plate velocity at ~6-km-depth in arrows representing the crustal deformation. ITO initial trench offset. Model parameters are given in Supplementary Table 2.

energy releasing[28] (Supplementary Fig. 2c). At about 2.76 Ma, as the continuous growth of the vertical shear zone, the through-lithosphere rupture has been developed, representing the maturation of mode-III VT (Supplementary Fig. 2a, b). After 2.76 Ma, the ceasing of VT growth (Supplementary Fig. 2b) and the stability of VT deformation rate (Supplementary Fig. 2c) indicate that the mature mode-III VT could propagate stably as a self-sustained geodynamic feature.

It is worth mentioning the morphology of the mature mode-III VT in both the vertical and horizontal directions. From a top view, the growth of tearing length stops once it cuts through the bending subducting lithosphere (Supplementary Fig. 2a). From a front view, the VT extends from the deep to shallow part of the shear region between two neighboring segments (Supplementary Fig. 2b). These two views indicate the morphology of a mature mode-III VT which is constrained within a narrow (few to ten kilometers wide) shear region between two neighboring segments (Supplementary Fig. 2a, b).

## Controlling factors of mode-III VT dynamics

Model results show the critical dependence of mode-III VT maturation and stable propagation on the trench offset geometry (Figs. 2 and 3). The critical trench offset for a mature mode-III VT changes by varying the subducting plate age and the strain weakening magnitude of the lithospheric mantle. The simulation results are summarized as three modes according to the propagation length of VT (Fig. 3): 1) the oblique subduction mode, where the intact plate subducts obliquely without the formation of mature mode-III VT; 2) the transition mode, where the mature mode-III VT can initiate but is sensitive to the shortening of trench offset length; and 3) the tearing mode, where the mode-III VT can propagate stably and the segmented subducting slabs can reach the mantle transition zone[29], representing the long-term stability of the mode-III VT.

Our experiments suggest that the trench geometry, offset by a transform boundary (Supplementary Fig. 1a), is required for the

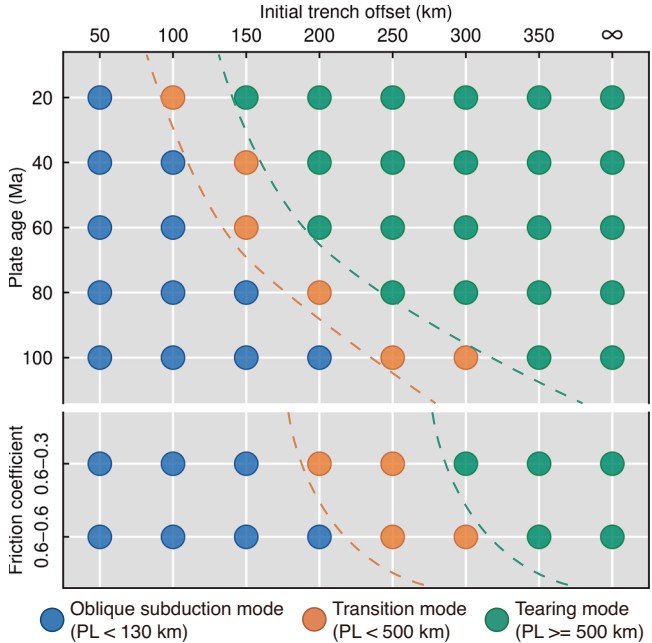

**Fig. 3 | Regime diagram of 3D experiment results.** The two groups correspond to different parameters (subducting plate age and strain weakening magnitude). In the plate age phase diagram (top group), the friction coefficient is 0.6-0, representing a strong strain weakening. In the strain weakening simulations (bottom group), plate ages are set to 40 Ma. Each of the circles refers to one of the numerical experiments, and their color indicates different modes: oblique subduction mode (blue), transition mode (orange), and tearing mode (green). The division of model results is according to the propagation length (PL) of mode-III VT (i.e., the length of the subducted slab). The 130 km distance, approximately corresponding to the length of the horizontal projection of the initial subduction interface, is regarded as a sign of mature mode-III VT (whether the mode-III VT could propagate completely into the plate interior). The 500 km distance indicates the subducting slab can reach the mantle transition zone between 410 km and 660 km discontinuities[29], representing a stably propagating mode-III VT. The orange and green dashed lines refer to the critical trench offsets for the maturation and for the long-term stability of VTs, respectively. The propagation length between 130 km and 500 km indicates a high sensitivity of mode-III VT to trench offset shortening processes.

maturation and long-term stability of mode-III VT within a homogenous subducting plate. For the reference model, with 40 Ma plate age and strong strain weakening, the 200 km trench offset enables the intact plate continuously be segmented into two subducting plate sections (Fig. 2a, Supplementary Fig. 3a–d). In contrast, mode-III VT did not appear in experiments with a 100 km trench offset (Fig. 2b, Supplementary Fig. 3e–h). The subducting slab is necking but not separated due to the insufficient stress concentration compared to the plate strength. The adjacent subducting slab ($Z > 250$km) underwent non-uniform rollback along the transform boundary, and ultimately, a curved trench and oblique subduction formed. This comparison indicates that trench offset geometry must provide sufficient stress amplification for the maturation of localized mode-III VT[4] (rather than distributed along-trench bending of the plate) around the outer subduction-transform corner. A sufficient trench offset (greater than the critical trench offset marked by orange dashed lines in Fig. 3) is also needed (i) to reduce the interaction between two neighboring subduction zones and (ii) to provide enough space for a complete mode-III deformation. This lateral non-uniform subduction assisted by mode-III VT is an alternative subduction mode compared to more uniform subduction associated with lateral slab bending along a curved trench (compare Fig. 2a and Fig. 2b).

The model sensitivity tests further demonstrate the critical control of the trench offset length on the mode-III VT. In the transition model,

with the same configuration as the reference model but a 150 km trench offset, the mature mode-III VT developed but ceased subsequently due to the gradual trench offset shortening with time (Supplementary Fig. 4a–d). This shortening is caused by the lateral crustal flow-induced retreat of the adjacent subducting slab ($Z > 250$km) and/or the shortening of the convex overriding plate ($Z < 250$km) under the convergent condition (Supplementary Fig. 4c, d). In the comparison model, a very small opposite velocity (1% of the subduction rate) on half of the overriding plate ($Z < 250$km) is preset (Supplementary Fig. 4e–h), which balanced the trench offset shortening to remain the trench offset constant at 150 km. The mode-III VT can thus propagate steadily for a long time. Furthermore, a larger opposite velocity (5% of the subduction rate) can induce the trench offset lengthening, which is also favorable to a self-sustained mode-III VT (Mod87 in Supplementary Table 2).

It is well understood that the oceanic plate has a colder thermal structure as it ages[30]. Our model results show that the role of the subducting plate age is twofold: (1) the colder thermal structure leads to a denser and rheologically stronger[31] lithosphere, increasing the plate bending and segmentation resistance. Thus, a higher stress amplification is needed to separate the older subducting plate; (2) the colder thermal structure implies a thicker lithosphere[30], which requires more space to accommodate a complete mode-III deformation (Fig. 2c). Therefore, as the model results summarized in Fig. 3, the older subducting plate requires a more significant trench offset to produce a mode-III VT.

As another crucial rheological control, the brittle/plastic strain weakening magnitude and dynamics determine the relative contribution of brittle and ductile deformation of the lithosphere (Methods). A strong strain weakening, with an intense and rapid lowering of the internal friction coefficient, dramatically facilitates strain localization. The mode-III VT emerges immediately when the subducting plate begins to bend (Supplementary Fig. 2), which performs like a brittle/plastic lithospheric fault. At the surface, the trench depth is relatively shallow and evenly distributed (Fig. 2a, c). In contrast, when the strain weakening is moderate or absent, the strain distribution tends to be spread rather than strongly localized. A more extended trench offset is needed for plate segmentation because of the tighter coupling between the two subducting segments (Fig. 3). The subducting slab warps and curves ahead of the propagation of mode-III VT (Fig. 2d), and the subsequent mode-III VT performs more like a ductile failure[14]. The magnitude of this pre-tearing flexure increases as strain weakening magnitude decreases (Supplementary Fig. 5). Consequently, a deeper and unevenly distributed trench depth develops in response to this slab ductile deformation (Fig. 2d).

**Comparison between mode-III VT and STEP**
Although the mode-III VT and the STEP develop in different geodynamic settings, our study indicates they share the same mechanisms and features. The major distinctions are the geodynamic settings at the surface and mantle flow in the deep (Supplementary Fig. 6). Mode-III VT separates two subducting plate segments with restricted rollback-induced flow between two segmented slabs. In contrast, STEP separates the subducting plate from the non-subducting plate[1,32] with the more extended rollback-induced flow. Nevertheless, these two processes have several similarities concerning their evolution and mechanisms. 1) They originate from the lateral non-uniform motion between the subducting plate and the neighboring plate. 2) The plate deformation mechanisms are the same. Both are mode-III shear deformation with the same evolution dynamics as described above. 3) They both develop at the outer subduction-transform corner and propagate in the opposite direction to subduction. 4) Both of them are strongly controlled by plate rheology[14]. 5) Both are self-sustained geodynamic features, as propagating stably once emerged[1]. By comparing the evolution of mode-III VT and STEP (Supplementary Fig. 6), the influence of the neighboring plate behavior on the propagation of

mode-III VT is imperceptible. Formally, STEP could be thus regarded as a mode-III VT with infinite trench offset length, which (according to our results) should enormously facilitate self-sustained tear propagation.

This formal unification makes it possible to use our trench offset geometry controlling mechanism to explain the findings in previous STEP studies. The STEP geometry can be regarded as an infinite trench offset length since the adjacent plate is not subducted, accounting for why the STEP can propagate stably once the STEP geometry exists[1]. Conversely, the lateral lithospheric weak zone in the neighboring plate, which can be treated as an embryonic subduction zone, makes the trench offset between the subducting and neighboring plates can be considered as absent. This explains why the existence of a lateral lithospheric weak zone leads to a whole oblique subduction zone through lateral subduction propagation rather than results in a trench sub-perpendicular plate segmentation via STEP[33].

## Originations of the trench offset geometry

The origination of the trench offset geometry is various. As in our models, the trench offset geometry can originate from the inversion of continental transform margins[34] (Fig. 1b). One natural case is the Tan–Lu Fault Zone, offsetting Dabie and Sulu Orogens, is regarded as a continental transform margin in the southern boundary of the North China Plate[35]. During the closing of the Paleo-Tethys Ocean and subsequent collision between the North China Plate and the Yangtze Plate, this trench offset geometry of the southern margin of the North China Plate induced the continuous mode-III VT within subducting the Paleo-Tethyan Oceanic Plate and the latter Yangtze Plate[35]. Moreover, studies about the mechanisms to collapse a passive margin proposed serval regions with high subduction initiation risk, of which the Argentine Basin and the U.S. East Coast (30°N–40°N) are the two regions containing continental transform margins and may evolve into active subduction in the next few tens of millions of years[36]. We speculate that the continental transform margins in these two regions (Fig. 1b) will perform as the initial trench offsets to form new mode-III VTs once the subduction is initiated. Endogenic plumes[31,37] or exogenic meteorite impacts[38] also could cause initial trench offsets by rupturing the integral lithosphere and inducing single-slab or multi-slab subduction[37]. However, we suggest them as potential origins because of the lack of natural examples in modern subduction systems. Another and more common origination of trench offset geometry is the inherited geometry from previous lateral non-uniform subduction induced by the subduction of lithospheric anomalies (buoyant terranes[19,20] or weak zones combined with lateral heterogeneity[25,39,40]). One natural instance is the subduction of the buoyant aseismic Caroline Island Ridge. The curvature, i.e., the trench offset geometry, of the southern Mariana trench is proposed as the result of the pinning of buoyant Caroline Island Ridge[22]. A mode-III VT is developed in response to this trench offset geometry to accommodate the lateral non-uniform subduction (Supplementary Fig. 7a, b). Another natural case is the subduction of the Meiji–Emperor–Hawaiian hotspot track. The subduction of this weak hotspot track facilitated the onset of non-uniform subduction between the Aleutian and Kamchatka trench since Neogene[10] (Supplementary Fig. 7c, d). After that, the offset between these two trenches has been growing gradually. It is worth noting that no lithospheric weak zone is consistent with the present positions of mode-III VT within the southern Mariana trench (-14.5°N)[22] and the Aleutian–Kamchatka trench (-173°E)[21] (Supplementary Fig. 7), indicating a potential transformation of the dominant control on the propagating mode-III VT from lithospheric anomalies to the trench offset geometry. The lithospheric anomalies-induced lateral non-uniform subduction results in the gradually offset trench, leading to the stress concentration and the localized plate strength weakening[41,42] around the outer subduction-transform corner. The formation and propagation of the mode-III VT, in turn, allows further non-uniform subducting to extend the lateral geometry difference[43]. Finally, once the growing

trench offset reaches the critical value (Fig. 3), the dominant control of the active mode-III VT would switch from the lithospheric anomalies to the extended trench offset geometry. The mode-III VT could thus propagate continuously into the intact subducting plate interior without the participation of lithospheric anomalies.

## Natural observations for mode-III VT

Mode-III VTs propagating within the intact subducting plate are broadly documented in modern subduction systems (Fig. 1b). They are characterized by the trench offset geometry with offsets from tens to hundreds of kilometers. With a small trench offset, the plate is necking at the outer trench-transform transition cusp but not segmented, which denotes an embryonic mode-III VT. This incipient mode-III VT with slight trench offset could be recognized in the northern Apennines[44,45] (Livorno–Sillaro Lineament, ~45 km offset, Supplementary Fig. 8a, b) and the western end of Hellenic subduction zone[13] (Kefalonia Transform Fault, ~100 km offset, Supplementary Fig. 8c, d). The further maturation of these incipient mode-III VTs requires the continuous growth of the trench offset by the assistance of lithospheric anomalies. Natural active mature mode-III VTs within the intact subducting plates can be recognized from the geophysical evidence in various subduction zones (Fig. 4a–c, Supplementary Fig. 7). The absence of intermediate-depth seismicity[46,47] (Fig. 4a–c) and reduction of seismic velocity anomalies[48–50] (Fig. 5a–c) around the outer subduction-transform corner indicate the existence of slab gaps between the subducted segments, which are suggested as the results of mature mode-III VTs[2]. The apparent trench offset geometry and the absence of lithospheric anomalies near the mode-III VTs indicate a trench offset geometry control.

Our sensitivity studies suggest that the variation of the trench offset length, caused by the lateral difference of trench migration, controls the propagation of mode-III VT, which is also documented in nature. The trench offset lengthening can be found at the western end of the Hellenic trench (Supplementary Fig. 8c, d), where the growth of the Kefalonia Transform Fault ensures the future maturation and continuous propagation of the embryonic mode-III VT[13]. In contrast, such as the geodynamic history of the Arabia-Eurasia convergence zone since the Eocene[51,52], the shortening of the transform boundary along the Eastern Caucasus-Western Iran boundary led to the halt of mode-III VT and the formation of the oblique convergence zone. Therefore, the fate of the modern active mode-III VTs can be roughly predicted by considering the present trench migration velocities[53]. The relatively stationary Solomon trench combined with a highly retreating Vanuatu trench shows a trench offset shortening in this region (Fig. 4a). A future ceasing of mode-III VT and a whole oblique subduction zone is predicted here. Conversely, the mode-III VT in the southeastern Aegean should be sustained in the future because of the rapidly retreating Hellenic trench (Fig. 4b).

The magnitude of brittle/plastic strain weakening, determining the relative contribution of brittle and ductile deformation, remarkably affects the slab deformation (Fig. 5) and the surface elevation (Fig. 4). Our model results with intense strain weakening show a mode-III VT performing like a brittle fault with less slab flexure (Fig. 2a) and an evenly distributed trench depth (Fig. 4b), which is in accordance with the investigations of the Solomon-Vanuatu trench[12] (Figs. 4a and 5a), Aleutian–Kamchatka trench[10,21] (Supplementary Fig. 7c), and Tonga trench[54]. In contrast, in the southeastern Aegean[11] and the northern end of Lesser Antilles[55,56], the subducting slabs flex significantly previous to mode-III VTs (Fig. 5b, c). As a result, the relatively deep parts of trenches, i.e., Rhodes Basin[57] (-4,485 m) and Puerto Rico trench[58] (-8,378 ± 5 m), have developed (Fig. 4b, c). These features may be attributed to the more intense slab ductile deformation in front of the mode-III VT, consistent with our models with moderate or without strain weakening (Figs. 4e, f and 5e, f).

In summary, we found that the Mode-III VT of subducting plates is a self-sustained process controlled by trench geometry and plate

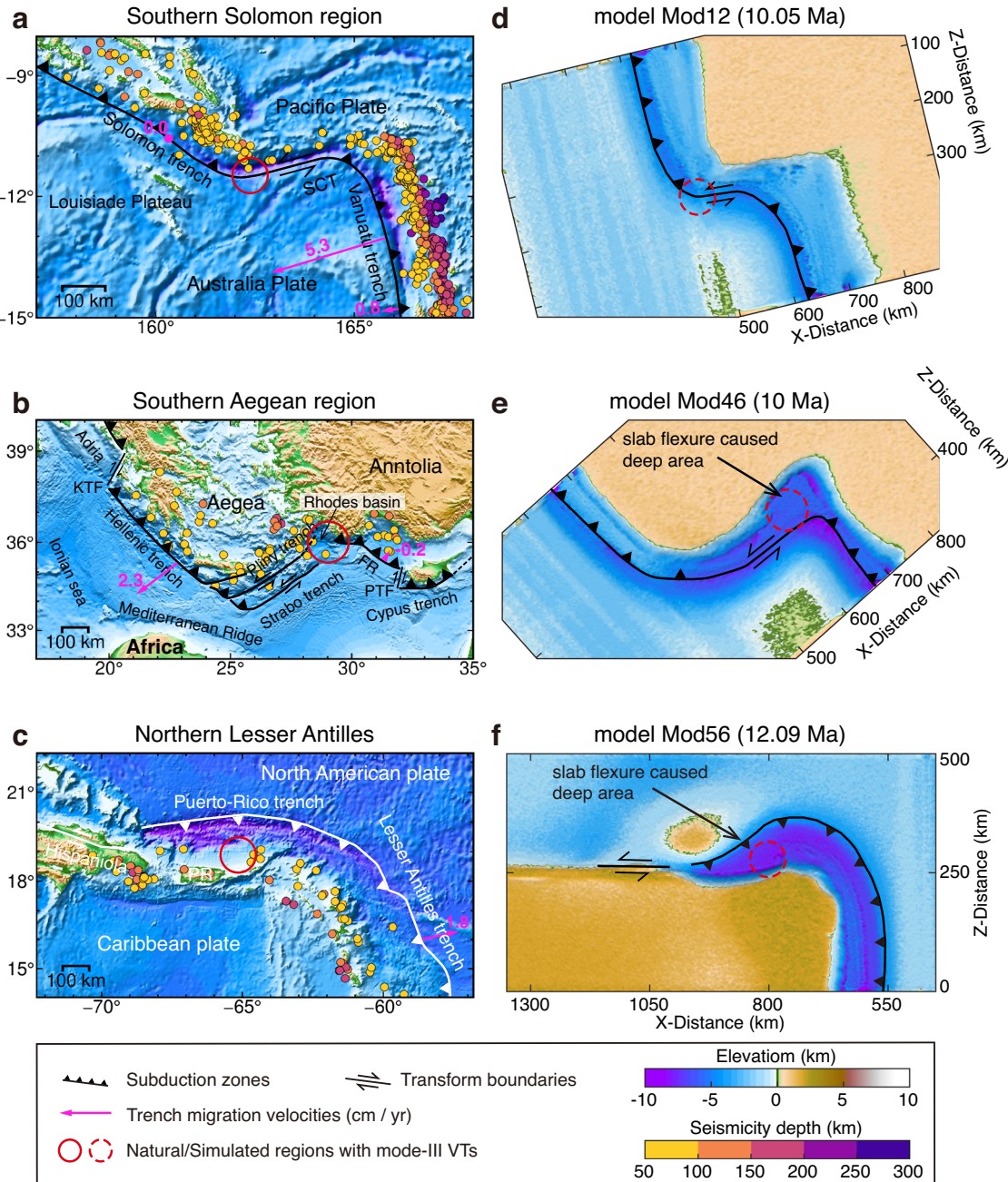

**Fig. 4 | Surface comparison of natural cases with numerical models.** Tectonics, intermediate-depth seismicity, topographic maps, and locations of mode-III VTs of Southern Solomon region[12,76] (**a**), Southern Aegean region[11,13] (**b**), and Northern Lesser Antilles[55,56] (**c**). The topography data is from ETOPO1 arc-minute Global Relief Model[75]. Seismicity data (Mw ≥ 5) is from the Global CMT catalog[46,47]. Trench migration velocities are from Schellart et al. (2007)[53]. SCT San Cristobal Trough, KTF Kefalonia Transform Fault, FR Florence Rise, PTF Paphos Transform Fault, PR Puerto Rico. Model parameters for **d**–**f** are given in Supplementary Table 2.

rheology. Mode-III VT performs as a lithospheric narrow shear zone that can propagate stably once matured if the condition remains the same. Trench offset geometry is an essential physical control of the long-term stability of mode-III VT within the intact subducting plate, as it can trigger stress concentration and further lithospheric weakening around the outer subduction-transform corner. The critical trench offset for a mature mode-III VT is strongly affected by the subducting plate rheology, i.e., the plate age and brittle/plastic strain weakening magnitude of the lithospheric mantle. Besides, the magnitude of brittle/plastic strain weakening can also significantly influence the slab deformation and the corresponding surface elevation by determining the relative contribution of brittle and ductile deformation. Our findings provide a new insight into the evolution of vertical tearing and

would contribute to the ongoing debate over the dynamics and physical controls of plate segmentation.

## Methods
### Numerical approach
The numerical experiments were conducted with the 3D thermo-mechanical code I3ELVIS, which combines a finite-difference method and a marker-in-cell technique[59,60]. The code solves mass, momentum, and energy conservation equations on a staggered Eulerian grid frame. Physical properties are transported via mobile Lagrangian markers that move based on the velocity field interpolated from the fixed Eulerian grid. Non-Newtonian viscous-plastic rheologies for different lithologies are taken in the model (Supplementary Table 3), which also accounts for

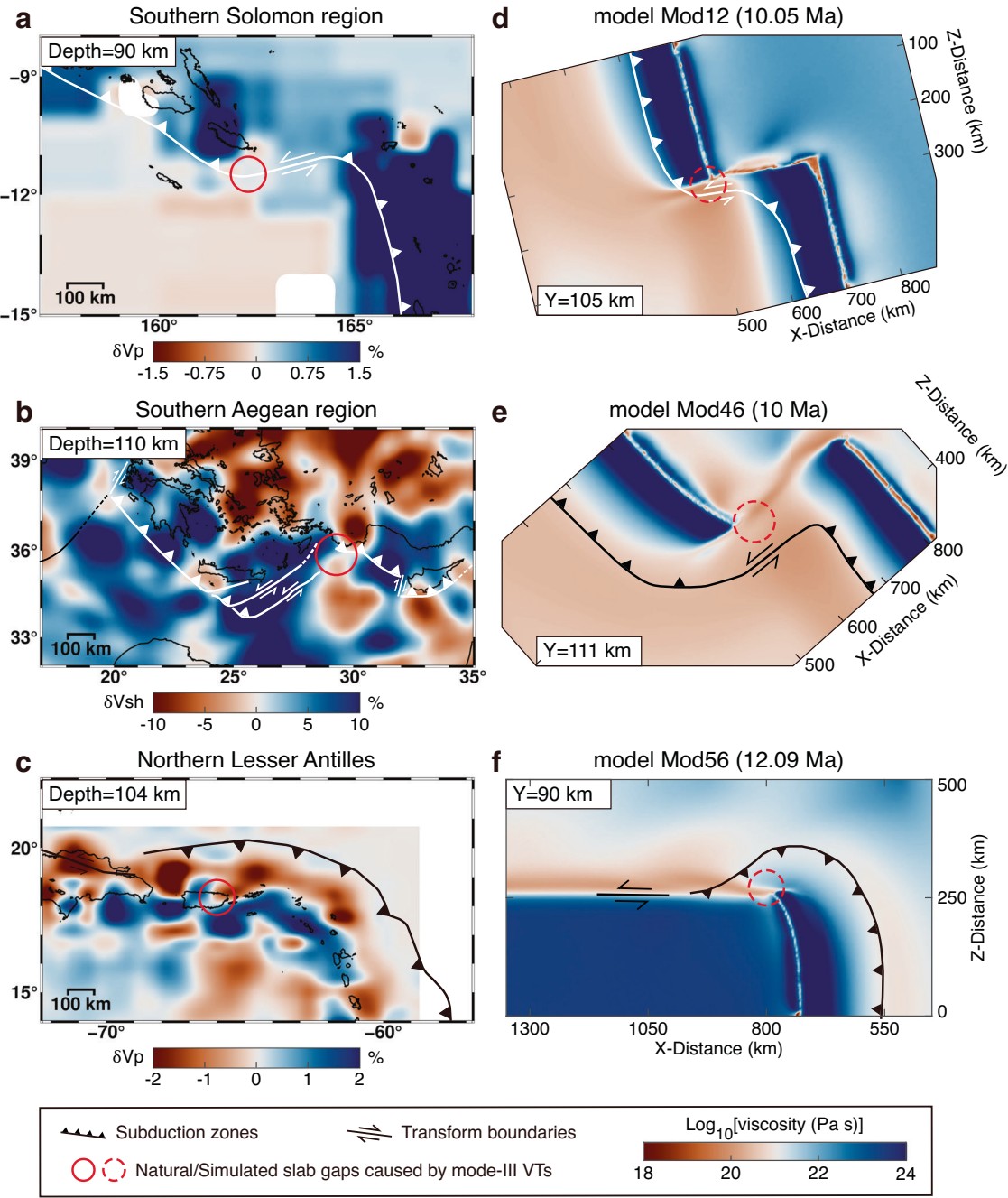

**Fig. 5 | Deep comparison of natural cases with numerical models.** Tectonics and tomographic images of Southern Solomon region (**a**), Southern Aegean region (**b**), and Northern Lesser Antilles (**c**). The distribution of seismic velocity anomalies in **a**–**c** is based on the tomography models of Amaru[48], Fichtner et al.[49], and Harris et al.[50], respectively. Model parameters for **d**–**f** are given in Supplementary Table 2.

adiabatic, radiogenic, latent, and frictional internal heating sources[31]. Detailed method description is provided in refs. 59,60.

**Numerical model configuration and boundary condition**

The initial model domain is 1500 km × 780 km × 500 km and is resolved by a regular rectangular grid of 405 × 261 × 181 Eulerian nodes with a resolution of 3.71 km × 3 km × 2.78 km in x, y, and z-direction, respectively. Over 158 million Lagrangian markers are randomly distributed in the whole computational domain.

The initial setup of the reference model (Mod 12) is shown in Supplementary Fig. 1. The typical model domain initially consists of a homogenous subducting oceanic plate and an overriding continental plate with different configurations and geotherms. In the

continental domain, a 35-km-thick continental crust, composed of 20 km upper crust and 15 km lower crust, lays on the 85 km lithospheric mantle and the subjacent 640 km asthenospheric mantle. The 8-km-thick oceanic crust comprises 4 km basalt and 4 km gabbro. The lower boundary of the oceanic lithosphere is equivalent to the depth of the 1400 K isotherm. A 20-km-thick 'sticky air' layer with low density (1 kg/m³) and viscosity (10^18 Pa s) is employed to implement a free surface condition at the top of the model box[61]. A trench offset geometry is prescribed, which corresponds to the transform section of the continental margin subjected to subduction initiation. In mature subduction zones, similar offsets can also form by other processes (e.g., by mantle plumes[31], meteorite impacts[38], or lithospheric anomalies[19,20,25,39,40]) and will represent the amount of

previous lateral non-uniform subduction. Two ~15 km thick weak zones with ~30° dip angles are defined between the subducting and overriding lithospheric mantle to simulate the initial subduction interfaces. The physical properties of rock materials are shown in Supplementary Table 3.

Free slip velocity boundary conditions are defined on all except the left ($X = 0$ km) and bottom ($Y = 780$ km) sides. A constant normal velocity is imposed on the left boundary to simulate continuous convergence. In some models, a constant normal velocity is also imposed at the right boundary ($Z = 1500$ km) to investigate the influence of plate velocities partitioning (Supplementary Table 2). A mass-conservative permeable boundary condition is imposed along the bottom boundary[62]. This infinity-like external boundary implies that a free-slip condition is satisfied at 297 km below the lower boundary of the model box. This external boundary condition allows the global conservation of mass in the computational domain.

The initial thermal structure of the oceanic plate with a prescribed plate age is computed according to the half-space cooling model[30], with 273 K at the surface and the mantle potential temperature of 1573 K. An initial adiabatic thermal gradient of 0.5 K/km is defined in the asthenosphere. The thermal boundary conditions are 273 K at the top and zero horizontal heat flux on all vertical boundaries. An infinity-like external boundary condition with a constant temperature is applied at the lower thermal boundary, ~297 km below the model base, which allows both temperatures and vertical heat fluxes to vary along the permeable box lower boundary[62].

## Rheology

The visco-plastic rheology of rocks is determined by experimentally determined flow laws (Supplementary Table 3), and the strength of rock materials is implemented through the evaluation of the effective viscosity, where the ductile viscosity and plastic viscosity are calculated separately. The ductile creep viscosity $\eta_{\mathrm{ductile}}$, which depends on pressure, temperature, composition, and strain rate invariant, is defined as a function of diffusion $\eta_{\mathrm{diff}}$ and dislocation $\eta_{\mathrm{disl}}$ creep:

$$\frac{1}{\eta_{\mathrm{ductile}}} = \frac{1}{\eta_{\mathrm{diff}}} + \frac{1}{\eta_{\mathrm{disl}}} \tag{1}$$

where $\eta_{\mathrm{diff}}$ and $\eta_{\mathrm{disl}}$ are computed by using linear Newtonian diffusion creep and power-law dislocation creep, respectively:

$$\eta_{\mathrm{diff}} = \frac{A_D}{2\sigma_{\mathrm{cr}}^{n-1}} \exp\left(\frac{E+PV}{RT}\right) \tag{2}$$

$$\eta_{\mathrm{disl}} = \frac{A_D^{\frac{1}{n}}}{2} \exp\left(\frac{E+PV}{nRT}\right) \dot{\varepsilon}_{\mathrm{II}}^{\frac{1}{n}-1} \tag{3}$$

where $P$ is the pressure, $T$ is the temperature (in kelvin), $R$ is the gas constant, $\sigma_{\mathrm{cr}}$ is the diffusion-dislocation creep transition stress[30], and $\dot{\varepsilon}_{\mathrm{II}} = \sqrt{\frac{1}{2}(\dot{\varepsilon}_{ij})^2}$ is the second invariant of the strain rate. The following parameters are experimentally determined material constant. $A_D$ is the pre-exponential factor, $E$ is the activation energy, $V$ is the activation volume, and $n$ is the stress exponent of the viscous creep.

The plastic (brittle) rheology $\eta_{\mathrm{plastic}}$ is implemented through the Drucker–Prager yield criterion, which is calculated as

$$\sigma_{\mathrm{yield}} = C_0 + P\varphi \tag{4}$$

$$\eta_{\mathrm{plastic}} = \frac{\sigma_{\mathrm{yield}}}{2\dot{\varepsilon}_{\mathrm{II}}} \tag{5}$$

Where $\sigma_{\mathrm{yield}}$ is the yield stress, $C_0$ is the material cohesion at $P = 0$, $P$ is the dynamic pressure, and $\varphi$ is the internal friction coefficient.

Although the physical mechanisms and conditions of lithospheric weakening are not fully understood, several processes have been suggested to account for the decreasing of the strength of deforming lithosphere, such as serpentinization[63], recrystallization[64], fluid-rock interaction caused by the fluid infiltration[65], self-localizing thermal runaway (shear heating)[66], coseismic dynamic weakening and grain-size reduction[67], microscale cavitation[68]. In particular, laboratory experiments have demonstrated that the friction coefficient is rock-type independent and determined that the friction coefficients of intact rocks are between 0.6–0.85[69]. However, within the seismogenic zone, the friction coefficients of minerals populating fault gouges varied from 0.8 to 0.07[70]. To represent the variety of possible weakening mechanisms in our model, a simplified empirical brittle/plastic strain-weakening approach is used, which facilitates strain localization by decreasing the internal friction coefficient of rocks with increasing strain. Such a simplified empirical strain weakening approach is widely used in geodynamic models of lithospheric deformation[71,72].

The brittle/plastic strain weakening is implemented by varying the strain weakening parameters, where the internal friction coefficient decreases in a linear manner with increasing brittle/plastic strain.

$$\varphi = \varphi_0, \text{ when } \gamma \leq \gamma_0 \tag{6a}$$

$$\varphi = \varphi_0 - (\varphi_1 - \varphi_0)\frac{\gamma - \gamma_0}{\gamma_1 - \gamma_0}, \text{ when } \gamma_0 < \gamma < \gamma_1 \tag{6b}$$

$$\varphi = \varphi_1, \text{ when } \gamma \geq \gamma_1 \tag{6c}$$

$$= \int \sqrt{\frac{1}{2}(\dot{\varepsilon}_{ij(\mathrm{plastic})})^2}\, dt \tag{7}$$

where $\varphi$ is the internal friction coefficient ($\varphi_0$ and $\varphi_1$ are the initial and final internal friction coefficient, Supplementary Table 2), $\gamma$ is the cumulative brittle/plastic strain ($\gamma_0 = 0$ and $\gamma_1 = 0.5$ are the lower and upper strain limits), $\dot{\varepsilon}_{ij(\mathrm{plastic})}$ is the plastic strain rate tensor, and $t$ is time (s). It should be pointed out that the strain weakening limit $\gamma_0$ and $\gamma_1$ are linearly scaled with the grid step[73], which ensures that the weakening scales with the absolute amount of material displacements.

Finally, the effective viscosity $\eta_{\mathrm{eff}}$ is calculated by combining ductile rheology with brittle/plastic rheology.

$$\eta_{\mathrm{eff}} = \min\left(\eta_{\mathrm{ductile}}, \eta_{\mathrm{plastic}}\right) \tag{8}$$

It should be mentioned that this large-scale viscous-plastic rheological model of the lithosphere is simplified, and the rock elasticity as well as ductile damage processes[74] are not taken into account.

## Minor sensitive parameter

We further studied the effects of convergence rate (2 cm/yr and 8 cm/yr) and plate velocities partitioning (overriding-plate driven retreat and slab-pull driven retreat). The convergence rate affects the model results by changing the duration of trench offset shortening. The plate velocities partitioning affects the model results by determining the subduction dip (Supplementary Fig. 9). One exception is the very young subducting plate (20 Ma) in the slab-pull driven retreat model (Mod73 in Supplementary Table 2), in which the slab broke off rather than subducted along a mode-III VT. It may be the consequence of a warm and weak lithosphere: on the one hand, the negative buoyancy of very young plates is insufficient to provide the necessary slab pulling force[31]; on the other hand, the resistance to mode-III VT is higher than

the resistance to break-off[1]. In general, these two factors influence the model evolution by affecting the propagation instead deciding the formation of mode-III VT. Both seem to play a minor role in the evolution of mode-III VT, and their effects on the model results are not very significant (Supplementary Table 2).

### Resolution sensitivity test

We tested the effects of model resolution along the direction of tearing propagation (*x*-direction) with both higher (2.60 km) and lower (7.08 km) resolutions compared to the reference resolution (3.71 km). The resolution experiments are conducted on models with intense as well as without brittle/plastic strain weakening (Supplementary Table 2). In models with intense strain weakening, we found some subtle differences in the propagation length of mode-III VT. However, the overall regime and behavior of the model do not change with different resolutions. For models without strain weakening, the required critical trench offsets decrease with higher resolution. A possible reason is that the stress amplification in the brittle part is more efficient at higher resolution because of narrower faults, which are always 1-2 cells wide. Although the critical trench offsets varied in different resolutions, the similar plate behaviors (i.e., slab flexure, uneven distribution of trench depth) still exist. As a consequence, our resolution tests show that the reference resolution is sufficient to investigate the most plate and tearing behaviors.

## Data availability

Due to the very large size of the data, the numerical results that support the findings of this study are available from the corresponding author upon request.

## Code availability

Researchers interested in using the I3ELVIS code should contact T. V. Gerya (taras.gerya@erdw.ethz.ch).

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

## Acknowledgements

This study was supported by the Second Tibetan Plateau Scientific Expedition and Research (2019QZKK0708) and the National Natural Science Foundation of China (U22B6002). The simulations were performed on the ETH-Zurich Euler and Leonhard clusters and on the Sunway TaihuLight System of the National Supercomputing Center in Wuxi. The open-source software ParaView (http://www.paraview.org) and Generic Mapping Tools were used for 3D and 2D visualizations, respectively. Vik colorscale from http://www.fabiocrameri.ch was used for 2D visualizations except topographic maps.

## Author contributions

Y.G.C. designed the study, performed the numerical experiments, interpreted the results, and wrote the manuscript. H.L.C. contributed to the model interpretation and visualization, and wrote the manuscript. M.Q.L. designed the study and contributed to the model tests and interpretation. T.V.G. provided the original code and wrote the manuscript. All authors discussed the results, problems and methods, interpreted the data, and reviewed the manuscript.

## Competing interests

The authors declare no competing interests.
