## [Peer Review File · Nature Communications]

REVIEWERS' COMMENTS

Reviewer #1 (Remarks to the Author):

This manuscript has been considerably improved since first submitted and is now much easier to follow. The authors have largely responded to my detailed criticisms but they did choose not to rewrite the paper to be more punchy. That is, perhaps OK now that the target journal gives scope for more details but I still think that if the opening words of the paper are "Mode-III (out-of-plane shear) vertical tearing (VT) ... " then the audience might be a bit limited.

I do appreciate the time the authors have taken to clarify the other points that I made in regard to the interaction of the slabs with the mantle flow that runs through the tear. We may differ on our interpretation of that, but I don't see this as a reason to hold up the paper. Let others read it to form their own opinions.

1 **Reviewers' comments**

2

3 Reviewer #1 (Remarks to the Author):

4

5 This manuscript has been considerably improved since first submitted and is now much
6 easier to follow. The authors have largely responded to my detailed criticisms but they
7 did choose not to rewrite the paper to be more punchy. That is, perhaps OK now that
8 the target journal gives scope for more details but I still think that if the opening words
9 of the paper are "Mode-III (out-of-plane shear) vertical tearing (VT) ... " then the
10 audience might be a bit limited.

11

12 I do appreciate the time the authors have taken to clarify the other points that I made in
13 regard to the interaction of the slabs with the mantle flow that runs through the tear. We
14 may differ on our interpretation of that, but I don't see this as a reason to hold up the
15 paper. Let others read it to form their own opinions.

16

17 **Response to the reviewers' comments**

18

19 **Reviewer #1**

20 1. This manuscript has been considerably improved since first submitted and is now
21 much easier to follow. The authors have largely responded to my detailed criticisms
22 but they did choose not to rewrite the paper to be more punchy. That is, perhaps OK
23 now that the target journal gives scope for more details but I still think that if the
24 opening words of the paper are "Mode-III (out-of-plane shear) vertical tearing
25 (VT) ... " then the audience might be a bit limited.

26 **Response:**

27 **We thank all reviewers for their time and valuable comments which have greatly**
28 **contributed to enhancing the accessibility and clarity of our work. The acronym**
29 **“VT” and the uncommon term “mode-III” are removed from the abstract.**
30 **However, as shown in Figure 1b, both tension stress and shear stress can induce**
31 **vertical tearing of subducting plates. In order to accurately specify our research**
32 **subject, we used an alternative expression like “shear stress-controlled vertical**
33 **tearing” (L20).**

34

35 2. I do appreciate the time the authors have taken to clarify the other points that I made
36 in regard to the interaction of the slabs with the mantle flow that runs through the
37 tear. We may differ on our interpretation of that, but I don't see this as a reason to
38 hold up the paper. Let others read it to form their own opinions.

39 **Response:**

40 **We would like to express our gratitude for the reviewer's understanding of our**
41 **different interpretation. We also appreciate the reviewers' acknowledgement of**
42 **the improvements we have made.**